# Comment on Wu et al. Baveno VII Criteria Is an Accurate Risk Stratification Tool to Predict High-Risk Varices Requiring Intervention and Hepatic Events in Patients with Advanced Hepatocellular Carcinoma. *Cancers* 2023, *15*, 2480

**DOI:** 10.3390/cancers16030661

**Published:** 2024-02-04

**Authors:** Manon Allaire, Dominique Thabut

**Affiliations:** 1AP-HP Sorbonne Université, Hôpital Universitaire Pitié-Salpêtrière, Service d’Hépato-Gastroentérologie, 75013 Paris, France; dominique.thabut@aphp.fr; 2INSERM UMR 1138, Centre de Recherche des Cordeliers, 75006 Paris, France; 3Sorbonne Université, INSERM, Centre de Recherche Saint-Antoine (CRSA), Institute of Cardiometabolism and Nutrition (ICAN), 75012 Paris, France

We read with great interest the original research conducted by Wu et al., which examines the utility of the Baveno VII criteria in the context of excluding varices needing treatment (VNT) in patients with hepatocellular carcinoma (HCC) [1]. While the study addresses a significant clinical concern, we have identified several critical issues regarding the results and conclusions put forth by the authors, besides semantics, i.e., the fact that Baveno VII criteria aim to rule out/rule in clinically significant portal hypertension (CSPH) in order to indicate carvedilol if needed, whereas Baveno VI criteria are devoted to excluding varices needing treatment (VNT) [2,3].

Firstly, the inclusion of 32% of the cohort comprising Child-Pugh B patients is noteworthy. Considering that the applicability of the Baveno VI criteria to rule out VNT is limited to patients with compensated cirrhosis, the exclusion of these patients from the analysis is crucial for an accurate assessment [2]. Moreover, the significant preponderance of patients with hepatitis B (70%) in the study cohort raises concerns about the extension of these results to metabolic dysfunction-associated steatotic liver disease (MASLD)-related HCC, a major cause of HCC in Western countries.

Secondly, we observed methodological discrepancies in the calculation of the missing VNT. Rather than calculating the VNT specifically among patients meeting the Favorable Baveno VI criteria (7/45, 17%), the authors calculated the VNT rate based on the general population (7/200, 3%). This oversight leads to a substantial difference in the interpretation of the data, indicating that the Baveno VI criteria may not be suitable for HCC patients, as already demonstrated by our team [4]. Moreover, we would also recommend a subgroup analysis according to the presence and extension of vascular invasion as it might impact the results.

Furthermore, while the study presents compelling data on the occurrence of liver decompensation events during the follow-up period, we express doubts about the inclusion of HCC progression as a liver-related event, particularly given the majority of patients falling under BCLC C (62%). Considering the strong association between HCC progression and liver decompensation due to disease advancement, definitive conclusions regarding the relationship to portal hypertension remain elusive.

Therefore, we recommend the implementation of more comprehensive investigations to determine the feasibility of employing both Baveno VI and VII criteria in the assessment of HCC patients. Further research in this area is essential to enhancing the understanding and management of HCC-related portal hypertension [5].

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
