# Peer review of "Comment on Wu et al. Baveno VII Criteria Is an Accurate Risk Stratification Tool to Predict High-Risk Varices Requiring Intervention and Hepatic Events in Patients with Advanced Hepatocellular Carcinoma. Cancers 2023, 15, 2480"

_cancers, 2024, doi:10.3390/cancers16030661_

Round 1
Reviewer 1 Report
Comments and Suggestions for Authors
This commentary is an interesting contibution to the debate about the applicability of the Baveno criteria in the setting of advanced HCC. I have some comments:
1. “Moreover, the significant preponderance of patients with hepatitis B (70%) in the study cohort raises concerns about the representativeness of the findings for the global HCC population”.
The authors recruited patients from their geographical area, i.e Asia. First, the authors would not have had ways to collect a different population without distorting data. Second, most clinical trials for unresectable HCC recruited populations with similar characteristics (including the Imbrave-150 trial of atezo/bev) and their representativity has never been questioned. Third, from an epidemilogical point-of-viewm, the so-called “global HCC population” is actually represented mostly by Asian and African patients with HBV-related HCC.
This reviewer understands that the authors probably meant that the findingsby Wu et al might not be extended to MASLD-related HCC automatically, as MASLD-related portal hypertension probably recognises peculariar mechanisms. Still, this sentence (in its present form) might be perceived as unnecessarly harsh and Western-centric. Therefore I suggest to better explain this concept.
2) Rather than calculating the VNT specifically among patients meeting the Favorable
Baveno VI criteria (7/45, 17%), the authors calculated the VNT rate based on the general
population (7/200, 3%)”. Here, the authors of this letter have a strong point. Both rates (7/45 and 7/200 are of interest for different reasons, as 7/200 represent the rate of VNT in the study population). Indeed, not providing the 7/45 rate was an oversight.
3) “Moreover, we would also recommend a subgroup analysis according to BCLC stages to ensure a more accurate assessment as the size and number of the tumor as well as the presence of vascular invasion might impact the results”. Actually, most patients were either in the BCLC-B or BCLC-C group. Therefore, the size and number of the tumour would not impact on the results, as they can not differentiated between B and C groups. The presence (and extension) of portal (and not generically vascular) invasion instead, is a matter of great interest, as a segmental portal vein thrombosis might probably have a different hemodynamic effect compared to a main branch or even a main trunk thrombosis. Even in this case, the extension of the thrombosis would not impact on the BCLC classification. Therefore, I would mention subgroup analysis but I would not tie these analyses to the BCLC classification automatically.
Author Response
Dear Editors and Reviewers,
We would like to thank you for your constructive comments that helped to significantly improve our manuscript entitled " omment on Wu et al. Baveno VII Criteria Is an Accurate Risk Stratification Tool to Predict High-Risk Varices Requiring In-tervention and Hepatic Events in Patients with Advanced Hepatocellular Carcinoma. Cancers 2023, 15, 2480” (Manuscript ID: cancers-2699591). As you will see below, we are providing a point-by-point response where we have addressed all the concerns that were raised and we have modified the manuscript accordingly.
In order to assist in the process, we included three writing modes in this document: black-bold for the original editors and reviewer’s comments as included in the decision letter, black-regular for our replies to reviewer’s comments and yellow underlined for changes made to the original version of the manuscript.
We greatly appreciate the opportunity to submit this revised manuscript and we hope it is now suitable for publication in your journal.
Sincerely,
Manon Allaire
- “Moreover, the significant preponderance of patients with hepatitis B (70%) in the study cohort raises concerns about the representativeness of the findings for the global HCC population”.
The authors recruited patients from their geographical area, i.e Asia. First, the authors would not have had ways to collect a different population without distorting data. Second, most clinical trials for unresectable HCC recruited populations with similar characteristics (including the Imbrave-150 trial of atezo/bev) and their representativity has never been questioned. Third, from an epidemilogical point-of-viewm, the so-called “global HCC population” is actually represented mostly by Asian and African patients with HBV-related HCC.
This reviewer understands that the authors probably meant that the findingsby Wu et al might not be extended to MASLD-related HCC automatically, as MASLD-related portal hypertension probably recognises peculariar mechanisms. Still, this sentence (in its present form) might be perceived as unnecessarly harsh and Western-centric. Therefore I suggest to better explain this concept.
Answer: we thank the reviewer for his comment and we modified the text accordingly
Moreover, the significant preponderance of patients with hepatitis B (70%) in the study cohort raises concerns about the extension of these results to metabolic dysfunction-associated steatotic liver disease (MASLD)-related HCC, a major cause of HCC in western countries.
2) Rather than calculating the VNT specifically among patients meeting the Favorable
Baveno VI criteria (7/45, 17%), the authors calculated the VNT rate based on the general
population (7/200, 3%)”. Here, the authors of this letter have a strong point. Both rates (7/45 and 7/200 are of interest for different reasons, as 7/200 represent the rate of VNT in the study population). Indeed, not providing the 7/45 rate was an oversight.
Answer: we thank the reviewer for his comment.
3) “Moreover, we would also recommend a subgroup analysis according to BCLC stages to ensure a more accurate assessment as the size and number of the tumor as well as the presence of vascular invasion might impact the results”. Actually, most patients were either in the BCLC-B or BCLC-C group. Therefore, the size and number of the tumour would not impact on the results, as they can not differentiated between B and C groups. The presence (and extension) of portal (and not generically vascular) invasion instead, is a matter of great interest, as a segmental portal vein thrombosis might probably have a different hemodynamic effect compared to a main branch or even a main trunk thrombosis. Even in this case, the extension of the thrombosis would not impact on the BCLC classification. Therefore, I would mention subgroup analysis but I would not tie these analyses to the BCLC classification automatically.
Answer: we thank the reviewer for his comment and we modified the text accordingly
Moreover, we would also recommend a subgroup analysis according to the presence and extension of vascular invasion as it mighst impact the results.